# A conceptual and computational framework for modeling the complex, adaptive dynamics of epidemics: The case of the SARS-CoV-2 pandemic in Mexico

Christopher R. Stephens[1,2], Juan Pablo Gutiérrez [ID][3]*

1 C3—Centro de Ciencias de la Complejidad, Universidad Nacional Autónoma de México, Mexico City, Mexico, 2 Instituto de Ciencias Nucleares, Universidad Nacional Autónoma de México, Mexico City, Mexico, 3 Center for Policy, Population and Health Research, School of Medicine, Universidad Nacional Autónoma de México, Mexico City, Mexico

* jpgutierrez@unam.mx

## Abstract

In the quest to ensure adequate preparedness for health emergencies caused by infectious disease pandemics, there is a need for tools that can address the myriad relevant questions related to the spread and trajectory of pandemics. A hybrid intelligence model that combines human and artificial intelligence may provide a viable solution, as it can process data from models that comprehensively integrate contextual and direct factors, effectively mimicking the social processes surrounding transmission while incorporating human interpretation to enhance our understanding of pandemics. Using data from the COVID-19 pandemic, we demonstrate the implementation of this approach with the publically available EpI-PUMA (*Ep*idemiological *I*ntelligence Platform for the Universidad Nacional Autónoma de México ("PUMA")) project and platforms, where a user may create their own hybrid intelligence Bayesian classifier models for a range of epidemiological indicators of interest. EPI-Puma integrates data from various public sources (including the national registry of SARS-CoV-2 cases, census data, poverty indicators, climate data and data related to atmospheric contaminants), enabling the deployment of models that predict a range of relevant outcomes. The main criteria for the data included was its coverage (at least at the municipality level) and availability (public data). EPI-Puma was able to identify both the differential predictive value of the different sets of factors related to the epidemic path and well as anticipate with a high probability the path of the pandemic (typical areas under the ROC curve for the associated classifiers being 0.8–0.9).

**Data availability statement:** The code for EpI-PUMA is available in https://github.com/chilam-lab/epipuma1.front.git.

**Funding:** We are grateful for financial support from DGAPA-PAPIIT project IV100520. the funders had no role in study design, data collection and analysis, decision to publish, or preparation of the manuscript.

**Competing interests:** The authors have declared that no competing interests exist.

## 1. Introduction

To comprehensively understand the behavior of epidemics, particularly those caused by communicable diseases like the recent COVID-19 pandemic, it is crucial to consider the context in which the transmission process occurs. This context has two crucial features: its high degree of multifactoriality/multicausality and its adaptability, which, together, characterize epidemics as Complex Adaptive Systems (CAS) [1,2].

While traditional models, such as the Susceptible-Infectious-Recovered (SIR) model, have been valuable, they address neither of these important features. Additionally, they may overestimate transmission dynamics, leading to challenges in controlling infectious diseases [3]. The epidemiological triad emphasizes the relationship between disease-causing agents, susceptible hosts, and the environment, highlighting both direct and indirect transmission routes [4]. Understanding the early growth dynamics of epidemics is essential for refining transmission models and improving disease forecasts [5].

In the case of infectious diseases, such as COVID-19, both individual behaviors and societal structures play a significant role in shaping the course of epidemics [6,7]. Models that consider the complexity of these mechanisms are essential for developing effective preparedness and response strategies. Mathematical and computer models have been instrumental in understanding the spread of infectious diseases and guiding containment measures [8]. By devising transmission models that capture epidemic growth profiles more flexibly, it is possible to enhance model fit, estimate key transmission parameters accurately, and improve epidemic forecasts [9].

As epidemics are the emergent outcome of a large number of stakeholder decisions at all levels, and it is exceedingly difficult to obtain data associated with these decisions, theoretical, as opposed to data-driven, models can and should be considered. Agent-based models have become feasible due to advancements in infectious disease modeling and computational power, allowing for a more detailed representation of disease transmission dynamics [10]. These models can help interpret randomized controlled trials and assess interventions to control pathogen transmission effectively [11]. Leveraging epidemiological models can aid in forecasting disease events and understanding the repercussions of mass gatherings on pandemics [12]. Similarly, game-theoretic models have also been considered [13,14].

Developing a conceptual model that considers the complexity of infectious disease transmission is crucial for estimating the level of incidence, route, and speed of transmission accurately. By integrating various factors such as individual behaviors, societal structures, and environmental influences, researchers can enhance preparedness and response strategies for future epidemics.

In this article, we propose a hybrid intelligence framework that considers the complexity in defining the route and speed of transmission of an infectious disease pandemic, exemplified by the case of the recent COVID-19 pandemic.

## 2. Conceptual framework

The chief motivation for the proposed framework is that an analytical approach to studying epidemics should be capable of addressing a broad range of pertinent

questions about different aspects of an epidemic. It should also be capable of leveraging existing data from a wide range of sources and disciplines that represent the complexity of the social and environmental interactions in which epidemics occur in order to answer these questions.

It is important to recognize that the progression of an epidemic is primarily shaped by the decisions made by stakeholders at all levels, influenced by their specific preferences, contexts and structural constraints. This includes individual choices, such as whether to wear a mask or wash hands. It also encompasses the decisions of healthcare professionals regarding, for example, when to intubate a patient or prescribe a particular medication, as well as the choices made by public health authorities, such as whether to implement a lockdown measure. All these decisions are constrained by social, economic and political determinants that affect the options available to each stakeholder.

Thus, the macro-dynamics of an epidemic is an emergent consequence of a vast number of micro-events associated with these decisions, and the context that shape them, at multiple levels. Crucially, every one of these decisions is at least partially based on a prediction and, consequently, a prediction model.

In spite of the promise and current interest in Artificial Intelligence (AI) and Machine Learning (ML), these decisions almost exclusively use "Human Intelligence" (HI)-based prediction models. Besides the intrinsic and numerous biases associated with human decision making, HI is not capable of incorporating the high degree of multifactoriality/multicausality that is a dominating feature of a CAS, such as an epidemic.

It is also difficult to objectively analyze the process of social determinants affecting health behaviors because each individual is inherently part of that process. Moreover, the vast amount of information that could and should be used to improve predictions and decisions is digital, and not readily accessible to HI. Equally, however, no AI algorithm is currently capable of exhibiting or unravelling the vast array of complex, causal relations that exist between the many factors that influence the progression of an epidemic.

Neither is there a capacity in AI to understand the constraints that govern whether an intervention is feasible or not. It is therefore to Hybrid Intelligence – an optimal combination of HI and AI – that we must look to in the future in the quest to predict and mitigate the evolution of epidemics, such as the SARS-Cov-2 pandemic [15].

In developing a model for an epidemic, two requirements are: i) that the model should lead to a better understanding of the factors and mechanisms that influence the progression of the epidemic, and ii) that it should be predictive [16]. In terms of predictability, it is essential to consider the broad range of relevant questions that may be of interest to different stakeholders, ensuring that the approach is capable of addressing their specific concerns. In other words, each stakeholder may have interest in a particular variable that they wish to predict, and a set of other variables to be incorporated as potentially actionable predictors.

## 2.1. Ecological versus epidemiological models

A basic division of variables is into ecological "where" variables, associated with a given spatial region, and epidemiological "who" variables, associated with a given population. Which perspective is adopted – ecological versus epidemiological – depends on the question of interest – is it a "who" versus a "where" question? and also on the nature of the data that is used to answer the question. For instance, those municipalities most likely to experience an increase in disease cases in the next month is a "where" question, whereas those demographic groups most likely to benefit in a reduction in mortality through a vaccination program is a "who" question. Similarly, publicly available socio-economic or socio-demographic data from a census is "where" data, while clinical data on the health status of an individual is "who" data.

Available information will often encompass both the "who" and "where" perspectives. For instance, from public health sources we may know the (anonymized) individuals that have tested positive for presence of SARS-Cov-2 - a "who" variable. If we have information about their location, we can count how many infected people there are in a certain place – a "where" variable. Similarly, when individual characteristics are unknown, aggregating information based on the characteristics of places can provide insights into transmission dynamics [17]. In this case, a "where" variable, such as the average

income in a census unit, can be used to characterize an individual that lives in that unit and distinguish them from individuals that live in units with a different average income.

In both ecological and epidemiological perspectives, a modeling unit must be determined. Generally, this modelling unit is implicit and associated with the scale of the decisions taken by the stakeholder. In the ecological context, for instance, decisions taken at the federal, state, municipality, or city level have their own intrinsic unit. However, a non-political division, such as partitioning a spatial region into a set of uniform spatial cells could also be used [18]. In the epidemiological context, a natural unit is the individual. However, most public health decisions must be taken at the population level, such as associated with a group with a certain demographic or medical characteristic.

In either case there exists an ensemble of modelling units, wherein variables can, in principle, be defined at any scale between the entire ensemble and an individual unit. Crucially, if one can model at the level of a single unit, one can aggregate to model at any more coarse-grained level. On the other hand, modeling at the level of the entire ensemble will not permit any predictions to be made at a more fine-grained level. In other words, you can derive the "macro" from the "micro" but not vice versa. This distinction between micro and macro is fundamental, in that, as emphasized, epidemics are emergent outcomes of an enormous number of micro-events that are shaped by decision-making in specific contexts, so that any macro-analysis represents a model of these underlying, causal micro-events.

## 2.2. Bayesian probabilistic modelling

We propose that an appropriate mathematical framework in which to model epidemics, or other CAS, is using a ML-based Bayesian classifier approach, where the fundamental quantity of interest will be $P(C|\mathbf{X})$, the probability to be in the class C, given conditioning factors $\mathbf{X} = (X_1, X_2,…,X_N)$. In the ecological perspective, C will be associated with a subset of spatial cells characterized by a particular criterion, such as those cells with the highest number of infections or deaths. In the epidemiological perspective it could represent a population, such as those men over 60 who have died, or diabetic women who have been hospitalized.

A Bayesian approach has multiple advantages. Firstly, through Bayes theorem it allows for the incorporation of potentially subjective information through a Bayesian prior $P(C)$. Secondly, it allows for the incorporation of new data $\mathbf{X}'$, such that a previous posterior, $P(C|\mathbf{X})$, becomes a new prior.

$$P(C|\mathbf{X}, \mathbf{X}') \ = \ P(\mathbf{X}'|\mathbf{X}, C)P(C|\mathbf{X})/P(\mathbf{X}'|\mathbf{X})$$

This feature is also very relevant for considering the concept of adaptability, as the probability distribution $P(C|\mathbf{X}(t))$, that describes a system at time t, adjusts to become a new distribution, $P(C|\mathbf{X}(t),\mathbf{X}'(t'))$, in the light of new information, $\mathbf{X}(t')$, at a later time t'. A third advantage is that it gives a very natural setting for considering issues of causality [19,20]. There is also evidence that HI itself is also Bayesian [21], so that an AI-based approach based on Bayesian classifiers can be more naturally synthesized into one, overall Hybrid Intelligence approach.

Although it is possible to formulate a Bayesian framework where C is a continuous variable, given that any continuous variable can be discretized, we will develop our conceptual model in the context where C is a discrete, categorical variable. We believe that this is also appropriate as HI is based on a categorization and weighting of a set of discrete factors.

To calculate $P(C|\mathbf{X})$, besides specifying what characteristic of the epidemic one wishes to model through the choice of a class C, two other essential components are: i) the choice of variables that will enter into $\mathbf{X}$, and ii) given a C and an $\mathbf{X}$, how to explicitly calculate $P(C|\mathbf{X})$.

## 2.3. Confronting multi-factoriality – determining predictability, causality and actionability

An important part of confronting the complexity of epidemics is to accommodate their high degree of multi-factoriality/multi-causality. From a prediction and decision-making perspective, although there are many component factors, $X_i$, that

make up **X**, it is important to be able to characterize each variable $X_i$ according to several criteria: i) the strength of the statistical association between C and $X_i$, as measured by its effect size and coverage; ii) its potential causality, with the causality potentially ranging from the most proximal causes to the most distal; iii) its degree of "actionability", i.e., to what degree it represents a factor that could be influenced by one or more interventions, or that it represents an intervention itself. It is also important to identify if it represents a "who" factor or a "where" factor, as this will strongly influence how it is incorporated into an explicit model.

The determination of predictability, causality and actionability for each variable is essential for creating better prediction models, and corresponding decisions, that are capable of mitigating the public health and social impacts of epidemics, independently of the degree to which HI or AI is used in the process. Our emphasis on the role of each $X_i$ stems from the fact that it is not sufficient to have a precise prediction of a property of an epidemic without some notion of "why", as it is the latter which is a crucial element in determining if an $X_i$ represents a potentially suitable target for an intervention, or an already implemented intervention itself.

For instance, if it is found that transmission rates are higher among a certain demographic group then it can be determined if an intervention may be targeted at that group. Moreover, if an intervention is implemented, the correlation, and possible causal link, between the intervention and the transmission rate may be computed.

The advantage of incorporating AI is its capacity to incorporate large amounts of digital data that can represent the multifactoriality of the problem. However, although it can also play an important role in helping distinguish between correlation and causation, it is HI, in the guise of knowledge, experience and expertise, that must also play a crucial role. Moreover, it is only HI, at least with the current state of AI, that is capable of determining the degree of actionability of a given factor or intervention.

## 2.4. Confronting multi-factoriality – determining an adequate set of predictors and their ontology

The identification of an initial set of factors **X** is a requirement, independently of whether an HI or AI perspective is adopted. From a Bayesian perspective, this can be thought of as a Bayesian prior. For instance, in a standard SIR-type model we assume implicitly that individual susceptibility factors play no role, and that transmission factors are the same for every individual. Similarly, an ecological model that includes climate factors, but not socio-demographic or socio-economic factors, is implicitly assuming that the latter have no effect [19].

As we will discuss presently, respecting the high degree of multifactoriality of epidemics presents a huge challenge in terms of data collection and integration. However, even with data in hand, the challenge is how to incorporate it into a prediction model and, furthermore, how to interpret its impact and make informed decisions. As mentioned, HI is not capable of incorporating a large number of factors into its prediction models, even less so when the data associated with those factors is digital.

On the other hand, given the requirement of having to examine a large set of $X_i$ factor by factor, even using AI requires the application of HI to analyze the potential degree of causality and actionability of each factor. Presenting the outputs of an AI model as a large set of indicators and statistics confronts us with an analog of choice overload bias [22], especially when many factors have similar degrees of predictability. What is required in this case is an ontology, or taxonomy, of the factors that allows and facilitates a more efficient analysis.

## 2.5. Confronting multi-factoriality – determining an adequate set of predictors and their ontology for the SARS-Cov-2 pandemic

To illustrate the challenges of creating an ontology and categorizing and cataloging potential causes it is useful to consider a particular epidemic, such as the SARS-Cov-2 pandemic, although the majority of the listed factors would be relevant for the majority of epidemics.

First of all, it is important to note that multiple ontologies can be created associated with different ways in which variables may be labelled. For example, in the context of understanding the transmission dynamics of infectious diseases like SARS-CoV-2, variables can be categorized as those that influence susceptibility versus exposure. Within the category of susceptibility factors and distinct to the initial assumption of universal susceptibility, implicit in SIR-type models for instance, we may highlight the importance of individual health status, particularly immunological factors, in modulating susceptibility to infections. For example, older individuals, on average, tend to be more susceptible due to age-related changes in immunity [23].

The presence of other diseases also plays an important role, [24,25]. These are clearly "who" factors that naturally enter an epidemiological model. In this example, we can also see emerge the role of disciplinarity in the development of an ontology. Those with expertise in immunology will be distinct to those with expertise in chronic diseases or nutrition, or stress and depression, all of which, and with many others, can affect susceptibility.

Moreover, although it is natural to think of susceptibility as a "who" variable, there are many potential "where" variables, or "who" variables that are only available from spatial data as "where" variables, that affect it, and which can serve as proxies. For instance, in the former case, levels of environmental contaminants can affect disease susceptibility [26], while, in the latter, socio-economic status can also be an important correlate of disease susceptibility [27].

Equally, although we may think of exposure as more naturally represented as a "where" variable, there are many "who" variables that modulate it, many of which are behavioral factors. For instance, whether or not one uses a facemask, or follows other public health recommendations [28]. The process of exposure to pathogens is central to understanding transmission dynamics.

Decision-making related to exposure, influenced by cognitive biases, plays a crucial role. For example, people often overestimate their ability to adhere to mitigation measures, with cognitive biases being more pronounced in individuals with lower education or income levels [29]. The interplay between living conditions that facilitate exposure and cognitive biases can increase the risk of pathogen exposure and severity of resulting conditions [30]. Again, a full understanding of all relevant exposure factors, as with susceptibility, requires an interdisciplinary approach.

As emphasized, epidemics are emergent outcomes of numerous micro-events shaped by decision-making by distinct stakeholders and in specific contexts. The concept of behavior, in this context, involves calculating the probability of certain actions by stakeholders given specific circumstances. Bayesian perspectives help in understanding and predicting behaviors based on available information [31]. To comprehend transmission probabilistically, we can do so either from the epidemiological perspective, using a statistical ensemble of "people", or from the ecological perspective, considering an ensemble of "places".

The division into ecological versus epidemiological variables itself also imposes an ontology that has implications for how to interpret information from a model, and who has the most appropriate profile to do so. For example, although information about places, including socioeconomic and health conditions, communication routes, economic activities, and climatic factors, can help in understanding the context that influences individual decisions related to exposure and transmission, each of these categories of factors requires different knowledge bases and experience. However, it is only by focusing on all these contextual factors that researchers can gain a comprehensive understanding of the dynamics of infectious disease spread [32].

Considering the complex interplay between individual susceptibility, decision-making processes, cognitive biases, and contextual factors is crucial for predicting the spread of infectious diseases such as COVID-19 and developing effective strategies to mitigate them. By integrating these elements first into conceptual and then computational models, researchers can enhance preparedness and response efforts for future epidemics.

Although the above categorizations are eminently sensible in terms of developing a deeper understanding of the factors that affect the dynamics of an epidemic, they are such that each category cannot be appreciated without an interdisciplinary approach. However, it may be that a more "disciplinary" categorization is fruitful, given that a better understanding of

causality and actionability may be gained by analyzing in more depth a particular category of variables, while recognizing that it is only the integration of these different disciplinary perspectives that will capture the underlying complexity of an epidemic. As an example of a more disciplinary categorization, below we consider some categories that have been found to be useful in the Epi-PUMA project and platforms that will be discussed further below and the reasons why they were incorporated.

## 2.6. Socioeconomic conditions of the population

Socioeconomic status is relevant as it contains information about individuals' conditions, as well as about their context. In relation to the probability of transmission, as previously noted, an association has been documented between socioeconomic status and cognitive biases that can affect the probability of exposure to infectious diseases. Likewise, occupation, in the case of economically active persons, also affects the probability of exposure to infectious diseases. Precarious employment, particularly jobs lacking assured income, can impact the probability of exposure to infectious diseases [33]. It is crucial to consider employment conditions when assessing health risks and exposure probabilities in relation to socioeconomic status.

**2.6.1. Welfare conditions.** Factors such as population density, housing situation, and ventilation play crucial roles for disease transmission. Population density is closely linked to the likelihood of exposure, as higher densities increase the probability of contact with infected individuals, thereby raising the risk of transmission [34]. In overcrowded settings, prolonged contact can further elevate the risk of exposure to infectious diseases [35].

The housing situation of individuals also influences the probability of exposure, with overcrowded living conditions potentially facilitating the spread of pathogens due to prolonged close contact among residents [36]. Moreover, the level of ventilation in indoor spaces is a significant factor affecting disease transmission. Adequate ventilation can help reduce the concentration of infectious particles in the air, thereby lowering the risk of transmission [37].

**2.6.2. Economic activity.** Economic activity significantly influences exposure to infectious diseases. The nature of economic activities in a region is closely linked to employment conditions, impacting the risk of pathogen exposure. Service-based economies involve increased person-to-person contact, including with travelers, heightening disease transmission risks [38].

**2.6.3. Health conditions of the population.** Health conditions of the population play a critical role in determining the likelihood of infection and the severity of disease outcomes. The initial health status of individuals significantly affects their susceptibility to infections once exposed. Health problems can weaken the immune response, increasing the likelihood of infection and potentially leading to more severe cases [39]. Individuals with pre-existing non-communicable diseases, such as diabetes, hypertension, and obesity are particularly vulnerable to experiencing more severe infections. These chronic conditions can exacerbate the impact of infectious diseases, leading to worse health outcomes and increased risk of complications [39].

**2.6.4. Social cohesion.** Social cohesion, defined as the connectedness and sense of belonging within a community, plays a significant role in influencing the probability of exposure to infectious diseases. The type and size of people's social networks, representing the level of social cohesion, are associated with the likelihood of exposure. In large and heterogeneous networks, the probability of exposure increases as there is a higher chance that someone in the network may be a carrier, potentially exposing the entire network [40]. Conversely, small and closed networks reduce exposure risks by limiting contact with external sources of infection [40].

**2.6.5. Communication routes and connectivity.** Communication routes and connectivity are crucial factors in the transmission of infectious diseases between locations. The movement of individuals carrying pathogens between different areas can lead to the spread of infections. In environments with increased traffic and connectivity between locations, there is a higher likelihood of exposure in areas where the pathogen was not previously present. Places with greater connectivity are not only more susceptible to the arrival of pathogens but also experience increased transmission to other locations [41].

**2.6.6. Availability of health services.** The availability of health services is crucial in the detection, diagnosis, and care of individuals with infectious diseases. Health services help reduce exposure to others by providing timely information on carriers and ensuring infected individuals receive appropriate care, thus aiding in managing the spread of infections and improving health outcomes [42].

**2.6.7. Climate and orography.** Climate and orography play significant roles in influencing the transmission dynamics of infectious diseases. The geographical features of a location, such as its topography and climate, can impact the ease of accessing and leaving the site, thereby affecting the transmission of pathogens. Additionally, certain ecosystems may provide more favorable conditions for pathogens to thrive, influencing their transmission patterns [43–45].

**2.6.8. Government programs.** Government programs and responses play a crucial role in influencing public behavior and mitigating the spread of infectious diseases. During emergency situations, targeted government support can impact individuals' decisions and behaviors, potentially reducing mobility and exposure risks. General limitations on mobility, as well as support directed at economic sectors dependent on personal assistance services, can also influence exposure levels [46].

### 2.7. Confronting multi-factoriality: The computational challenge

As there is ample evidence that a host of different factors can influence the propagation of an epidemic, we may ask the question: why are they not all included in epidemiological models? In terms of an HI model, we have explained the barriers to incorporating a large number of variables based on digital data. In our ML-based Bayesian perspective they can be included by constructing a vector of attributes **X** that represents as complete a set as possible of the factors that are posited to influence the propagation of the epidemic. In the ecological perspective this means obtaining as much information as possible about a given place, while in the epidemiological perspective it means obtaining as much information as possible about a given person.

The first problem to consider then is for a given factor, $X_i$, where do we obtain data for that factor for a person or a place? What exists versus what is readily available? There are two important steps: i) the identification of potentially useful datasets; and ii) their integration into a data architecture that makes them available for modelling. For point i) the key word is "useful". If we restrict attention to the argument that two key requirements are to explain and predict an epidemic, then, without an objective criterion with which to define them, it is disciplinary bias that is an important component in deciding which factors to include and which not.

As mentioned, in Bayesian terms, this corresponds to a choice of Bayesian prior. If a set of variables **X´** is not included, then, implicitly, we are assuming that $P(C|\mathbf{X´}) = P(C)$. Furthermore, if we do not later collect data on **X´** and include it in the likelihood $P(\mathbf{X´}|C)$, then we cannot correct our prior using that data. Thus, as an example, if we thought of mobility factors **X** or socio-economic factors **X´** as being potentially predictive, then two models that calculated or measured $P(C|\mathbf{X})$ or $P(C|\mathbf{X´})$ would give us only marginalized information relative to $P(C|\mathbf{X},\mathbf{X´})$. To consider the latter we need a statistical ensemble, where for a unit in that ensemble we have information about both **X** and **X´**.

Fortunately, from an ecological perspective it is possible to integrate data from very different sources about a particular place. For example, it is possible to integrate census data and public health data for each census unit in a country. In principle, the same is true for a person. However, this is rarely if ever the case, as disciplinary studies focus on data for a specific population and a particular subset of variables, **X**. Another study carried out with respect to a set of variables **X´** is not commensurate with the first unless carried out on the same population. This distinction is important when we consider the integration of data in a data architecture to facilitate its inclusion in a model.

Another important data issue is the different possible data formats or resolutions of a given data source. In terms of different "format", there are different characterizations that have implications about how to handle data in terms of integration, both outside and, more particularly, inside a model. Some relevant categorizations of data are: quantitative versus

qualitative; nominal, ordinal, discrete or continuous; structured, semi-structured and unstructured; time series data, spatial data and text data. For a given data type, such as spatial data, there are also multiple data formats, as in raster, or vector, or whether it is 2D or 3D. Within these data types there are also different spatial resolutions. For instance, does a pixel in a raster represent an area 1km X 1km, or 2km X 2km etc.?

It is a significant challenge to integrate very different data types, formats and resolutions into one single representation $X$ that can be input into a prediction model $P(C|X)$. Below we will mention one explicit approach to this problem.

## 2.8. The role of explainable AI

There are many potential algorithmic representations of $P(C|X)$. In the case where $X$ is highly multifactorial, these have to be approximations, as the frequentist representation $P(C|X) = N(CX)/N(X)$, where $N(X)$ is the number of occurrences of the state $X$ in the ensemble and $N(CX)$ is the number of co-occurrences of the state $X$ and the class of interest C, is inadequate, as $N(CX)$, $N(X) = 0, 1$ when $X$ has high dimension. There are many potential representations of $P(C|X)$, from standard logistic regression to a sophisticated Deep Learning Neural Network.

Although, inevitably, there exist many technical differences, the most important in our context are: i) how good is the prediction? ii) how well does it explain? How well a given algorithm predicts can be determined in standard fashion: divide data into training and test sets, generate the model on the training set and validate it on the test set using one or more performance metrics. In principle, this offers an objective path to deciding what is the best algorithm. However, besides which class of ML algorithm will be used - logistic regression, recurrent neural network, random forest, decision tree, xg boost etc. - given that each algorithm is associated with a choice of parameters, there is also the question of what specific instance of a particular algorithm will be used.

Furthermore, there is the question of which features $X_i$ will be used. If there are N features, there are $2^N$ possible models covering each possible set of chosen features. As it is not feasible to check the performance of each possible model, either an algorithmic feature selection mechanism must be used [47], or the modeler must make a choice based on HI. Feature selection is crucial, as it is precisely the degree of predictability in the relations between the $X_i$ and C that will determine the performance of the model.

The question of explainability is more subjective than that of predictability. However, it is a vitally important element in the case of epidemics. As we have repeatedly emphasized, it is the nature of the $X_i$ themselves, and their relations with C, that represent both the possibility to understand the causes of the epidemic and its progression, as well as the impact of interventions that are used to mitigate its progression.

## 3. Computational framework

In this section we will present an explicit computational representation of our conceptual framework in the context of the EpI-PUMA (An *Ep*idemiological *I*ntelligence Platform for the UNAM ("PUMA")) project and platforms [48,49], designed and implemented for the SARS-Cov-2 pandemic in Mexico. We emphasize that this is just one among many potential computational realizations. However, we believe that any other realization to be successful must confront the conceptual and data challenges we have described above.

The principal data challenges are the identification and integration of potentially incommensurate data sets that can represent, at least partially, the high degree of multfactoriality/multicausality associated with the evolution in space and time of the COVID-19 epidemic, their subsequent incorporation into Bayesian classifier models and, finally, the visualization of these models for analysis and decision-making. In Fig 1 we see a diagrammatic representation of these processes. The first task was to identify a set of readily available data sources that could represent various potential risk factors/causes, as previously discussed.

As long as the etiological agent and the transmission mechanism are known - aspects that define the direct cause of the infection and allow us to know the infectious capacity of the agent-, there is a set of elements that affect, on the one

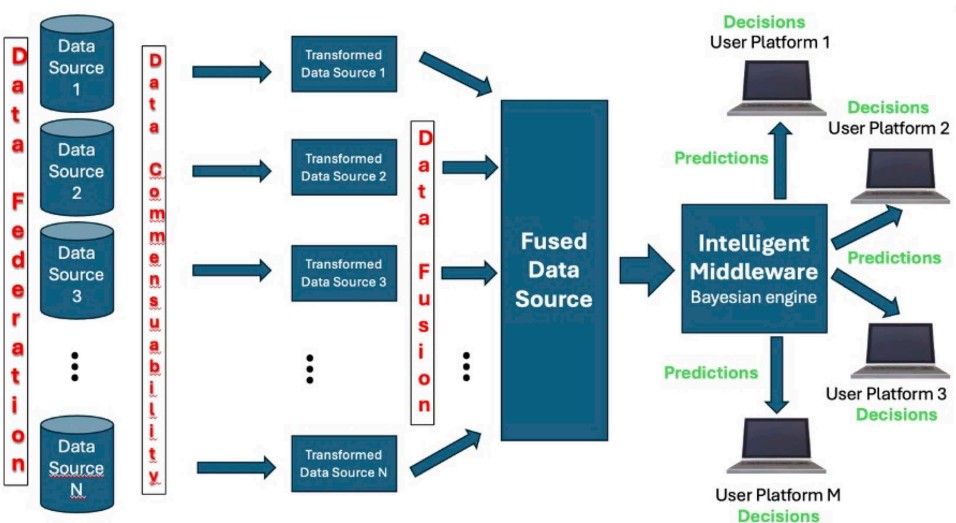

**Fig 1. Schematic representation of the data flows in the Epl-PUMA 1.0 platform.** Upon choosing a data field to be the class of interest, C, the Intelligent Middleware module implements the Bayesian classifier framework, using the Naive Bayes approximation to calculate a score proxy for $P(C|\mathbf{X})$ and a score $s(X_i)$ for each individual predictor. The results of the model can then be visualized as is shown in the following figures.

hand the probability of exposure (and therefore of infection), and on the other hand the severity of the infection once it has been contracted.

These elements include both those that characterize people in their social position and health background, as well as those related to mitigation measures and people's ability to follow them. As noted, identifying the pandemic pathway in a country or sub-national context requires characterizing how different elements facilitate or reduce both the likelihood of exposure and the severity of the disease.

With regard to the probability of exposure, once the virus is in circulation in the environment of the individuals, as has been explained in general terms previously, this can be affected by individual and family behaviors as well as by decisions exogenous to the individuals. The immediate living environment (the number of close contacts of individuals and the number of close contacts of close contacts) also plays a role. As a whole, both the decisions and the characteristics of the immediate environment occur within the framework of the individual's social position, which conditions or limits the scope of action of individuals.

The severity of the disease has been identified as being associated with previous health conditions, but it is also related to the initial dose of virus, which is influenced by the same elements mentioned in relation to exposure.

The decision-making process is framed by a set of cognitive biases that make it difficult to identify the alternatives that effectively reduce exposure, although they can be oriented with clear guidelines from the authorities.

This simple process is complex, involving, among others, the probabilities of exposure, the site of attack, the amount of virus (a virion or a high concentration of pathogen), and the clinical susceptibility (initial immune status).

With this in mind, the data sets chosen were the 2020 Mexican Census data (https://www.inegi.org.mx/programas/ccpv/2020/), poverty and social welfare data (https://www.coneval.org.mx/Medicion/FI/PMP/Paginas/Modulo-de-Condiciones-Socioeconomicas.aspx), intra- and intermunicipality labor mobility (derived from Mexican Census data), climate data from Worldclim [49] and data on atmospheric contaminants obtained from the Instituto de Ciencias de la Atmósfera y Cambio Climático of the National University of Mexico. Of course, we do not wish to suggest that these sources represent a final and definitive set – far from it. Research is ongoing to incorporate other relevant data sets.

In Fig 1, these, together form a data federation, where each data source consisting of data vectors $\mathbf{X}_\alpha$, $\mathbf{X}_\beta$ etc. can be consulted individually. However, to form an overall vector $\mathbf{X} = (\mathbf{X}_\alpha, \mathbf{X}_\beta,...)$ that combines data from each source it is necessary to perform a transformation step to bring each data source to the same spatial resolution. Once this is done the transformed data sources can be fused together.

### 3.1. EpI-PUMA 1.0 an ecological computational framework for the SARS-Cov-2 pandemic in Mexico

In the specific case of the COVID-19 epidemic, the EpI-PUMA 1.0 platform (https://epipuma10.c3.unam.mx/) was developed from an ecological perspective, with an approach based on considering the degree to which a given place offered conditions (context) that could be considered as favorable - "niche-like" - or unfavorable - "anti-niche-like" - for a particular epidemiological aspect of the epidemic. As an ecological model the system is based on an ensemble of spatial elements – Mexican municipalities – that form a partition of the spatial region of interest – the Mexican republic.

In terms of the logic of our conceptual framework, to generate Bayesian classifiers P(C|$\mathbf{X}$), the EpI-PUMA 1.0 platform involves choosing a class of interest C and a set of potential causes $\mathbf{X}$ (Fig 2). The platform then uses the Naive Bayes approximation to compute an approximation to P(C|$\mathbf{X}$) that relates each individual factor $X_i$ to C, quantifying both the degree of predictability of $X_i$ and its statistical significance (Fig 3). With P(C|$\mathbf{X}$) in hand it is possible to calculate for any element, α, of the spatial ensemble P(C(α)|$\mathbf{X}$(α)), the probability that in the cell α, given the multifactorial niche profile $\mathbf{X}$(α) of that cell, that the epidemiological variable of interest, C, takes the value C(α).

Given the Bayesian classifier nature of our framework, the epidemiological aspects of the pandemic are captured as "risk groups". For instance, those municipalities that will have the highest number of positive cases of COVID-19 in

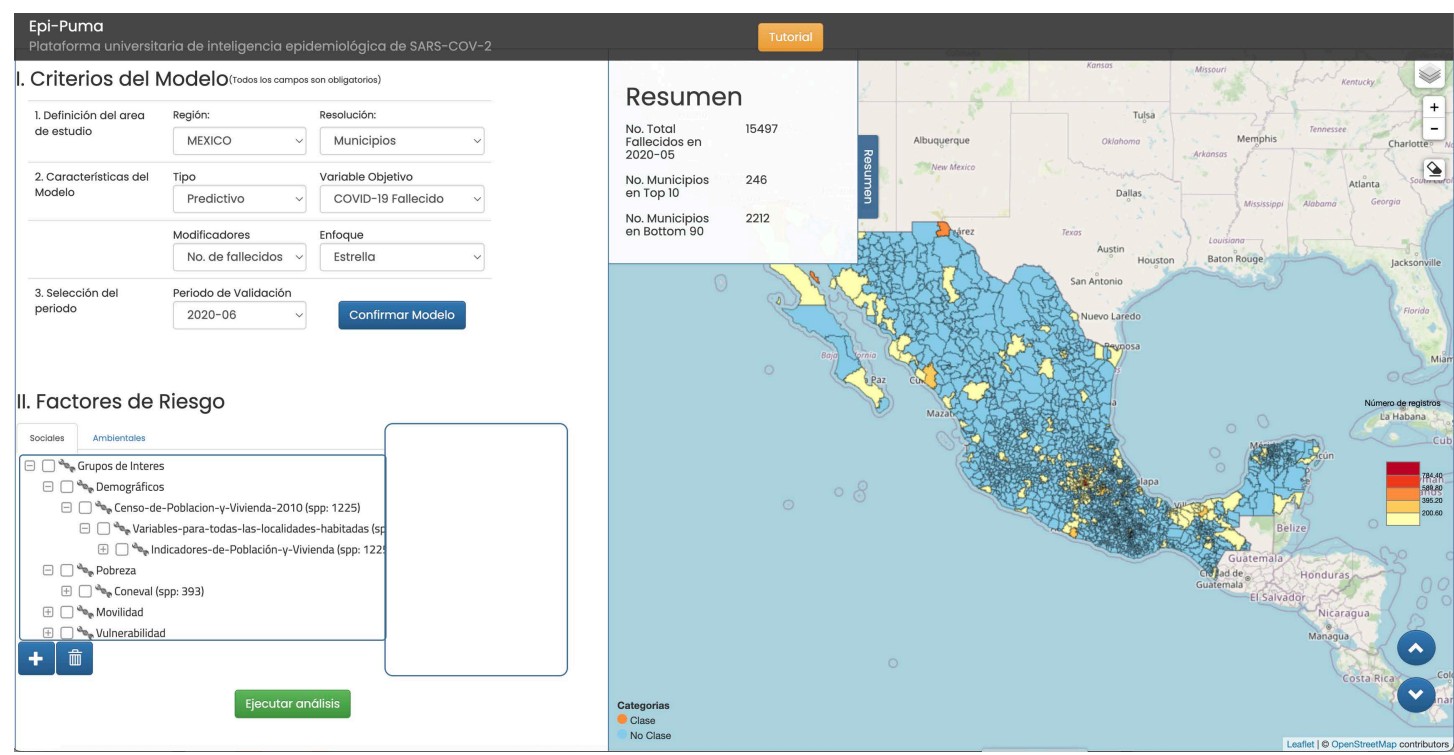

**Fig 2. Model configuration view of EpI-PUMA 1.0.** A choice of class C is made in "Criterios del Modelo". In this case the required prediction is the 10% of municipalities that are predicted to have the highest number of deaths in June 2020. The choice of predictors is made in "Factores de Riesgo" and the system computes an approximation to P(C|$\mathbf{X}$) on choosing "Ejecutar análisis". (The EpI-PUMA 1.0 mapping functionality uses the freely available Leaflet mapping scripts (https://leafletjs.com/) and OpenStreetMaps for data (https://www.openstreetmap.org/copyright)).

the next month, or the highest mortality, or those municipalities most likely to show a significant increase or decrease in deaths. Over 20 distinct epidemiological variables can be modelled as classes C [50].

The ontology used was chosen to respect the "disciplinary" origin of the data sources. However, we emphasize that the system allows for the rapid creation (on the order of seconds) of a Bayesian classifier-based prediction model using any combination whatsoever of variables from those available. By doing so it is possible to determine if, for example, climatic factors are more predictive than socio-economic factors, or if demographic factors are more predictive than mobility factors.

Note that EpI-PUMA 1.0 provides solutions for several of the computational challenges alluded to in section 2e). First, it assigns values to all variables to be used to each element of the same statistical ensemble of spatial units – in this case Mexican municipalities. Thus, raster (pixel) based variables from WorldClim are evaluated at a municipal level. Secondly, all ordinal variables are discretized so that every variable is representable as a set of categories wherein any given spatial unit can be assigned a set of binomial variables, including the class of interest.

The system shows the predictive performance of each variable group chosen (Fig 3) in terms of the model´s recall. In this case, using data on the reported number of deaths by municipality and month from the Mexican Ministry of Health, we chose as class C the 10% of municipalities that from May 2020 were predicted to have the highest number of deaths in June 2020. As the horizontal axis corresponds to a ranking in deciles of the municipalities according to P(C|**X**) for the given model, decile 10 will yield a recall of 10% if the corresponding model is not predictive at all.

In this case, the estimated models are highly predictive. Shown are 4 models corresponding to the distinct sets of risk factors chosen: Census-based socio-demographic and socio-economic factors (Covariable 1), mobility factors (Covariable 2), climatic factors (Covariable 3) and a model combining all factors together (Total). We can clearly see that climatic factors are much less predictive (recall 29% in the top decile) than mobility, or socio-demographic and socio-economic factors (recall 69% and 65% respectively in the top decile).

It is important to emphasize that the EpI-PUMA 1.0 platform allows users to generate their own predictive model based on their own choice of C and **X**. Thus, the model performance is principally determined by the predictability intrinsic to the relation between C and **X** rather than being a sensitive function of the ML algorithm being used.

Finally, as we have emphasized the need to be able to analyze the role of each variable separately, in Fig 4 we see how the system allows for such an analysis. Categoría (category), Fuente (source), Tipo de Dato (Data type) and Covariable are ontological descriptors. Pc is the null hypothesis probability for finding a municipality in the top 10% with the highest number of deaths in June 2020, while Pij is the probability to find a municipality associated with the shown covariable in the top 10%. Score is the weight of evidence of the covariable, and Epsilon is the result of a binomial test that measures the statistical significance of the difference between Pij and Pc, i.e., to what degree the observed distribution is in accord with the null hypothesis.

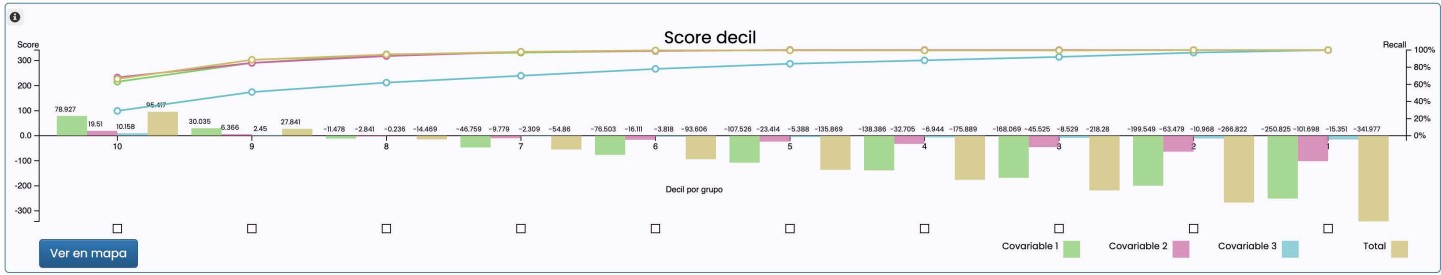

**Fig 3. Model performance view of EpI-PUMA 1.0 showing the results of four individual models according to the risk factors chosen.** Covariable 1 represents socio-demographic and socio-economic factors, Covariable 2, mobility factors, Covariable 3, climatic factors and Total represents a model that includes all the factors.

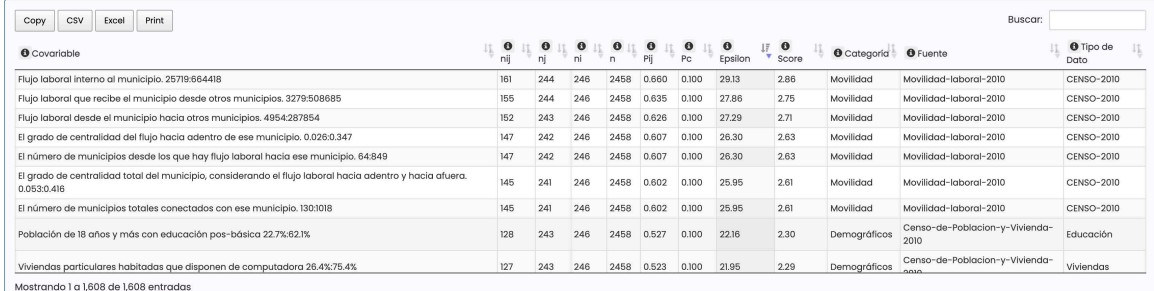

| Covariable | nij | nj | ni | n | Pij | Pc | Epsilon | Score | Categoría | Fuente | Tipo de Dato |
|---|---|---|---|---|---|---|---|---|---|---|---|
| Flujo laboral interno al municipio. 25719:664418 | 161 | 244 | 246 | 2458 | 0.660 | 0.100 | 29.13 | 2.86 | Movilidad | Movilidad-laboral-2010 | CENSO-2010 |
| Flujo laboral que recibe el municipio desde otros municipios. 3279:508685 | 155 | 244 | 246 | 2458 | 0.635 | 0.100 | 27.86 | 2.75 | Movilidad | Movilidad-laboral-2010 | CENSO-2010 |
| Flujo laboral desde el municipio hacia otros municipios. 4954:287854 | 152 | 243 | 246 | 2458 | 0.626 | 0.100 | 27.29 | 2.71 | Movilidad | Movilidad-laboral-2010 | CENSO-2010 |
| El grado de centralidad del flujo laboral hacia adentro de ese municipio. 0.026:0.347 | 147 | 242 | 246 | 2458 | 0.607 | 0.100 | 26.30 | 2.63 | Movilidad | Movilidad-laboral-2010 | CENSO-2010 |
| El número de municipios desde los que hay flujo laboral hacia ese municipio. 64:849 | 147 | 242 | 246 | 2458 | 0.607 | 0.100 | 26.30 | 2.63 | Movilidad | Movilidad-laboral-2010 | CENSO-2010 |
| El grado de centralidad total del municipio, considerando el flujo laboral hacia adentro y hacia afuera. 0.053:0.416 | 145 | 241 | 246 | 2458 | 0.602 | 0.100 | 25.95 | 2.61 | Movilidad | Movilidad-laboral-2010 | CENSO-2010 |
| El número de municipios totales conectados con ese municipio. 130:1018 | 145 | 241 | 246 | 2458 | 0.602 | 0.100 | 25.95 | 2.61 | Movilidad | Movilidad-laboral-2010 | CENSO-2010 |
| Población de 18 años y más con educación pos-básica 22.7%:62.1% | 128 | 243 | 246 | 2458 | 0.527 | 0.100 | 22.16 | 2.30 | Demográficos | Censo-de-Poblacion-y-Vivienda-2010 | Educación |
| Viviendas particulares habitadas que disponen de computadora 26.4%:75.4% | 127 | 243 | 246 | 2458 | 0.523 | 0.100 | 21.95 | 2.29 | Demográficos | Censo-de-Poblacion-y-Vivienda-2010 | Viviendas |

Mostrando 1 a 1,608 de 1,608 entradas

**Fig 4. Predictive and statistical significance analysis of the 1608 variables considered.** Thus, for example, the first entry (of 1608) shows that: thought of as a network of labor flows, the probability that a municipality that has the highest degree of direct intra-municipal labor flow has a probability of 0.660 of being in the top 10% of municipalities with the highest number of deaths in June 2020. Thus, of the 244 municipalities (nj) associated with that covariable, 161 also had the highest number of deaths in June 2020. This is extremely statistically significant in that Pij = 0.660 is 29.13 standard deviations of the binomial distribution distant from the null hypothesis Pc = 0.1. Similarly, the second entry shows the importance of inter-municipal labor flows.

As the principal goal of this paper is to present a conceptual framework for modeling epidemics as CAS, and the corresponding requirements of a computational framework, this discussion of EpI-PUMA 1.0 serves to demonstrate the feasibility of the presented framework in a concrete setting. A more detailed description of the system, the precise nature of the ML modelling algorithm used, and its results have been published elsewhere [48,49].

## 3.2. EpI-PUMA 2.0 an epidemiological computational framework for the SARS-Cov-2 pandemic in Mexico

As well as an ecological analysis, it is possible to make a concrete realization of the Bayesian classifier approach in the epidemiological context, where a population of people, as opposed to a population of spatial units, is considered. Indeed, many ML-based models of SARS-Cov-2 have been formulated. In EpI-PUMA 2.0 (https://epipuma20.c3.unam.mx/), this has been done using clinical states of persons, such as if they were hospitalized, or in an Intensive Care Unit, or intubated, or have died, as classes of interest. The ontology of predictors used is associated with different disciplinary, or stakeholder, categories. For instance, the socio-demographic and socio-economic characteristics of the person, either from knowledge of the person as an individual - "who" variables, such as age and gender – or via "where" variables - associated with the place they are resident, using census data. A category of health status variables is used that covers, principally, comorbidities. Another category used is that of clinical variables which, for example, permit the user to see the probability of death conditioned on the intubated state of the person and compare it, as a risk factor, to others, such as being diabetic or being old etc.

All the different data sources in the EpI-PUMA platforms are incorporated into a modular data architecture and transformed into a format such that data from different sources and of different types can be included as predictors, **X**, or classes of interest, C, in the same classifier model that calculates P(C|**X**).

## 4. Conclusions and recommendations

The conceptual framework discussed above seeks to address four fundamental characteristics of an epidemic that need to be modelled: i) that there are many variables of interest, associated with different stakeholders, that need to be predicted; ii) that the evolution in space and time of these variables is highly multi-causal, with causes that span the "micro" to the "macro"; iii) that the relationships between these variables and their causes is dynamic and adaptive; iv) that decisions need to be taken to mitigate its consequences and that these decisions should be based on adequate prediction models.

We have argued that a Bayesian classifier system that calculates conditional probabilities, P(C|**X**), is an appropriate framework in which to model an epidemic, as it satisfies property i), through allowing for a wide choice of different classes

of interest, C, within the same modelling framework. It also satisfies requirement ii), in that it allows for the incorporation of a, potentially, large set of predictive features $\mathbf{X}$. It satisfies iii) in that Bayes theorem can be invoked to incorporate new information, and satisfies iv) in that the predictions $P(C|\mathbf{X})$ can be used to power the decisions of stakeholders, where the impact of an intervention can also be predicted by including it as another factor in $\mathbf{X}$.

To successfully operationalize this conceptual framework computationally, there are several important barriers to be addressed. First of all, the choice of an appropriate class of interest and predictors is not as simple as it sounds. Often, stakeholders, through bias or lack of information, focus on questions that are not necessarily the most important in terms of reduction of impact of the epidemic or, worse, focus on predictors that are either not actionable or not directly causal.

Secondly, given the highly complex, multifactorial nature of epidemics, obtaining data sets of very different types and formats, and from quite different sources, presents a major challenge, especially in the case of the epidemiological perspective, where many data sets are not commensurable, such as having genetic data, but no mobility data, for one population, and vice versa for another. There are two approaches to this problem, one is to develop highly multifactorial sets for a given population in the first place, such as the UK Biobank initiative, while the other is to have government and science policy promote population "sharing", so that new experimental protocols can be carried out on populations for which there already exist data from previous protocols. The result is that instead of having data equivalent to distinct models for $P(C|\mathbf{X})$ and $P(C|\mathbf{X}')$, one can develop models for $P(C|\mathbf{X},\mathbf{X}')$. Furthermore, it is necessary that any mitigation measures adopted by stakeholders at different levels can be explicitly represented as digital data and fed into the prediction models, thereby allowing the possibility of predicting the future impact of the intervention and its optimization.

Thirdly, one must develop an explicit algorithmic representation of $P(C|\mathbf{X})$ that accounts for the potentially high degree of multifactoriality in $\mathbf{X}$ and is capable of incorporating variables of very different data types, formats and resolutions. Furthermore, in order to be accepted into the decisions of stakeholders it should yield explainable predictions. This puts constraints on which type of AI/ML is preferable to use and, given that it must be included in stakeholders´ decisions, it must allow for the integration of HI, i.e., a Hybrid Intelligence approach must be adopted. This approach must allow for the evaluation and categorization of a large set of features according to their predictability, their potential causality and their potential actionability

We have demonstrated that a system can be developed that combines all of the above features. EpI-PUMA makes it possible to generate reliable estimates of the evolution in space and time of multiple ecological and epidemiological characteristics of a communicable disease epidemic, as was shown in the case of COVID-19. The predictive capacity of the model crucially depends on the degree of predictability inherent in the data that is used to calculate $P(C|\mathbf{X})$. Once a dataset has been identified, transformed and incorporated into the EpI-PUMA architecture it can be used in the creation of multiple ML models along with other data sources. Moreover, the predictability, potential causality and actionability of any given set of variables can be compared and contrasted with the same from any other set, as was explicitly shown above for the case of climatic, demographic and mobility data. This ability to quickly construct multiple models that test different hypotheses is crucial, given that stakeholders have different interests as to which subsets of predictors are relevant, and which are possibly under their control.

Of course, although the EpI-PUMA platforms integrate many different datasets, by no means do we mean to imply that the set of predictors included constitute a full and final set. On the contrary, it is just a beginning. Several other datasets have been identified and are currently being included. For instance, although EpI-PUMA incorporates variables from social structure and context through census data, the social determination process itself is not directly observable in this data. However, it is modeled by incorporating most relevant data, allowing this process to be recognized in the estimation.

In the ongoing global efforts to enhance pandemic preparedness and strengthen the capacity to respond promptly and effectively when the next pandemic arises, the framework proposed here could play a valuable role. It emphasizes the necessity for a Hybrid Intelligence approach that can generate and interpret the results of the multifactorial models which reflect the complex, adaptive social processes in which epidemics occur. However, although we have shown the feasibility of the approach, we need to recognize the enormous challenges that must be overcome to implement it. The

delineation of these challenges may serve as a set of recommendations to different stakeholders, particularly for the public health authorities. First and foremost, it requires a recognition of the implications of the highly multifactorial nature of an epidemic. The fragmentation of relevant data across multiple government and non-government agencies means that we only see a very partial picture of what is happening. Partial both in the sense of incomplete and in the sense of biased according to the data collectors´ view as to what data is important. The collection and integration of relevant, multi-scale, multidisciplinary data is vital if we are to develop a more holistic view of an epidemic. The task of turning such data into predictions using ML models we believe is eminently feasible, as we hope we have shown. However, perhaps the most significant barrier is the incorporation of predictions that capture the complex, multifactorial nature of an epidemic into stakeholder workflows and decision cycles. Having to account for and organize information associated with potentially hundreds, or even thousands, of potentially direct and indirect causes, and use it to arrive at an optimal decision is something quite foreign to HI, which has evolved to recognize in most situations the relevance of only a few key features. This is why the development and application of suitable ontologies is important.

## Acknowledgments

We have benefited from the dedicated efforts of the whole Epi-PUMA team.

## Author contributions

**Conceptualization:** Christopher R. Stephens, Juan Pablo Gutierrez.

**Data curation:** Christopher R. Stephens.

**Writing – original draft:** Christopher R. Stephens, Juan Pablo Gutierrez.

**Writing – review & editing:** Christopher R. Stephens, Juan Pablo Gutierrez.

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
