## [Decision Letter · Decision Letter 0]

31 Jan 2025

PONE-D-24-43200A Conceptual and Computational Framework for Modeling the Complex, Adaptive Dynamics of Epidemics: the case of the SARS-CoV-2 pandemic in Mexico.PLOS ONE

Dear Dr. Gutierrez, Thank you for submitting your manuscript to PLOS ONE. After careful consideration, we feel that it has merit but does not fully meet PLOS ONE’s publication criteria as it currently stands. Therefore, we invite you to submit a revised version of the manuscript that addresses the points raised during the review process.  Please see below the reviewers' comments. *Please be aware that you reserve the right to disregard references suggested by reviewers if they are considered irrelevant or not pertinent to the content and focus of the paper. * Please submit your revised manuscript by Mar 17 2025 11:59PM. If you will need more time than this to complete your revisions, please reply to this message or contact the journal office at plosone@plos.org . Please include the following items when submitting your revised manuscript:

We look forward to receiving your revised manuscript.

Kind regards,

Youssef El Khatib, Ph.D.

Academic Editor

PLOS ONE

Journal Requirements:

“We are grateful for financial support from DGAPA-PAPIIT project IV100520. “

“We are grateful for financial support from DGAPA-PAPIIT project IV100520. We have also benefited from the dedicated efforts of the whole Epi-PUMA team.”

 “We are grateful for financial support from DGAPA-PAPIIT project IV100520. “

6. We note that [Figure 1] in your submission contain [map/satellite] images which may be copyrighted. All PLOS content is published under the Creative Commons Attribution License (CC BY 4.0), which means that the manuscript, images, and Supporting Information files will be freely available online, and any third party is permitted to access, download, copy, distribute, and use these materials in any way, even commercially, with proper attribution. For these reasons, we cannot publish previously copyrighted maps or satellite images created using proprietary data, such as Google software (Google Maps, Street View, and Earth). For more information, see our copyright guidelines: http://journals.plos.org/plosone/s/licenses-and-copyright.

Additional Editor Comments:

I recommend revising the manuscript based on reviewers' reports.

Reviewers' comments:

Reviewer's Responses to Questions

**Comments to the Author**

1. Is the manuscript technically sound, and do the data support the conclusions?

Reviewer #1: Yes

Reviewer #2: Yes

2. Has the statistical analysis been performed appropriately and rigorously? 

Reviewer #1: Yes

Reviewer #2: Yes

3. Have the authors made all data underlying the findings in their manuscript fully available?

Reviewer #1: Yes

Reviewer #2: Yes

4. Is the manuscript presented in an intelligible fashion and written in standard English?

Reviewer #1: Yes

Reviewer #2: No

5. Review Comments to the Author

Reviewer #1: Reviewer comment

Manuscript Number: PONE-D-24-43200

Manuscript Title: A Conceptual and Computational Framework for Modeling the Complex, Adaptive Dynamics of Epidemics: the case of the SARS-CoV-2 pandemic in Mexico

General comments:

The manuscript entitled “A Conceptual and Computational Framework for Modeling the Complex, Adaptive Dynamics of Epidemics: the Case of the SARS-CoV-2 Pandemic in Mexico” by Juan Pablo Gutierrez et al., demonstrates the implementation of a hybrid intelligence model that combines human and artificial intelligence to improve preparedness for the health emergencies like infectious diseases pandemics using data from the COVID-19 pandemic. The topic is interesting and the manuscript is written well. However, the manuscript is not deemed fit for publication in its current status and it needs a minor revision and improvement. I have a few comments that should be addressed:

Including a line number would have made it much easier to comment.

1. Abstract

“EPI-Puma integrates data from various sources…” Define the abbreviation “EPI-Puma” when it first appears in the manuscript. The sentence also mentions integrating data from various sources but doesn’t identify which sources are used, based on what justification the data are selected and how they are incorporated.

The abstract doesn’t provide how accurate the EPI-puma project and platform are in predicting the outcomes.

The abstract is overly brief and lacks sufficient details. Expanding it more on the aspect of methodology and implication is recommended.

The abstract needs to be revised and edited for improved clarity and simplicity.

2. Introduction: Well-crafted and clearly conveys the purpose. Nicely presented setting the stage for the rest of the content.

3. Conceptual and Computational Frameworks: The paragraphs under both topics are very elaborative and verbose (some are more than 20 lines), but it is overwhelming and makes it challenging to follow the flow of ideas. So, breaking it into smaller paragraphs and making it precise will enhance clarity and better understanding.

4. Conclusions and Recommendations:

- Should be number 5, not 4

- It is well-written

Reference: OK,

- Some of the references have a reference code and most don’t have it. For example, reference number 8, 23, 26, 30, 35, and 37. But most of the reference lacks it. So, make it uniform and correct it all according to the journal’s referencing style.

Reviewer #2: The present article focuses on adequate preparedness for health emergencies caused by global pandemics like COVID-19. The authors proposed a theoretical framework required for addressing many questions related to the transmission dynamics and disease trajectory of these pandemics. They introduced a hybrid intelligence model that combines human and artificial intelligence that may offer a viable solution by processing data from various sources. Eventually, it effectively mimics the social processes surrounding transmission while incorporating human interpretation to enhance our understanding of pandemics.

The topic addressed here is extremely interesting, and their simulation-backed results reported in the following part anyhow justify the strength of their model. At the same time, it also gives a solid reason to address four fundamental characteristics of an epidemic that need to be modelled by an Artificial Intelligence tool. Undoubtedly the authors dedicated themselves to performing a comprehensive analysis of the model they proposed.

However, after reviewing the manuscript (MS) is that the work in its present form still has several weaknesses. I believe, the manuscript is still lacking some crucially important points, which requires a thorough revision before getting accepted for publication in PLOS ONE.

To this end, I would like to ask the authors to append a few additional information, as listed below, that will drive to a substantial improvement to the following asking issues.

Comment #1.

The introduction part seems too short to describe some motivational facts behind this study. I would like to encourage the authors to make a robust discussion highlighting the influence of individual decision-making in uncertain environments. In addition, they may cite some articles showcasing the necessity of classical game models to trace out the opinion dynamics as well as its impact on public decision-making. Inline to that they may take help from the following textbooks/articles:

(i) Heidecke et al.; A mathematical model to assess the effectiveness of test-trace-isolate-and-quarantine under limited capacities, PLOS ONE, 2024.

(ii) Tanimoto Jun; Sociophysics Approach to Epidemics, Springer, 2021.

(iii) Kim et al.; Tracing and testing multiple generations of contacts to COVID-19 cases: cost-benefit trade-offs, ROYAL SOCIETY OPEN SCIENCE, 2022.

(iv) Tanimoto Jun; Fundamentals of Evolutionary Game Theory and its Applications, Springer, 2015.

(v) Maria Martcheva; An Introduction to Mathematical Epidemiology; Springer, 2015.

(vi) Ghosh et al.; A mathematical model for COVID-19 considering waning immunity, vaccination and control measures, Scientific reports, 2023.

(vii) Daems et al.; The Race for COVID-19 Vaccines: Accelerating Innovation, Fair Allocation and Distribution, Vaccines, 2022

Comment #2.

In your proposed model, you have mentioned the basic division of variables fall into two categories, namely ecological and epidemiological. Can you please justify which type of variable is more influential while modeling an epidemic like COVID-19?

Comment #3.

The authors are encouraged to add different types of decision-making protocols into the probabilistic model they used. I believe, it would be more interesting for the general audience of PLOS ONE. For example, they can cite the following recently published articles to catch things up while elaborating the updating protocols.

(i). Mansura et. al.; An in-silico game theoretic approach for health intervention efficacy assessment, Healthcare Analytics, Vol. 5, 100318, 2024.

(ii). Kulsum et. al.; Modeling and investigating the dilemma of early and delayed vaccination driven by the dynamics of imitation and aspiration, Chaos, Solitons and Fractals, Vol. 178, 114364, 2024.

(iii). Alam et al.; A Game-Theoretic Modeling Approach to Comprehend the Advantage of Dynamic Health Interventions in Limiting the Transmission of Multi-Strain Epidemics, Journal of Applied Mathematics and Physics 10 (12), 3700-3748, 2022.

Comment #4.

It may also improve their current MS if the figure captions would be made more self-contained. More precisely, one could also consider a sentence or two saying what is the central theme or message of each figure. Also, the authors are requested to add more visuals that can convey the central theme of the proposed model for general readers.

In this regard, the authors may cite the following works:

(i). Tatsukawa et. al.; Stochasticity of disease spreading derived from the microscopic simulation approach for various physical contact networks, Applied Mathematics and Computation, 431(80):127328, 2022.

(ii). Alam et al.; A Game-Theoretic Modeling Approach to Comprehend the Advantage of Dynamic Health Interventions in Limiting the Transmission of Multi-Strain Epidemics, Journal of Applied Mathematics and Physics 10 (12), 3700-3748, 2022.

Comment #5.

I would suggest the authors to extend their abstract more with the key results. As it is, the abstract is a little thin and does not quite convey the interesting results that follow in the main paper.

Besides as above, I have no further comments, and I expect a thoroughly revised version of this manuscript would be a great contribution for PLOS ONE.

6. PLOS authors have the option to publish the peer review history of their article (what does this mean? ). If published, this will include your full peer review and any attached files.

**Do you want your identity to be public for this peer review?** For information about this choice, including consent withdrawal, please see our Privacy Policy .

Reviewer #1: No

Reviewer #2: No

---

## [Author Response · Author response to Decision Letter 1]

20 Mar 2025

Thanks for your very valuable comments that we addressed in the revised version. Please see the attached file with answers and/or actions to each comment.

---

## [Editor Report · Decision Letter 1]

9 Apr 2025

A Conceptual and Computational Framework for Modeling the Complex, Adaptive Dynamics of Epidemics: the case of the SARS-CoV-2 pandemic in Mexico.

PONE-D-24-43200R1

Dear Dr. Juan Pablo Gutierrez,

We’re pleased to inform you that your manuscript has been judged scientifically suitable for publication and will be formally accepted for publication once it meets all outstanding technical requirements.

Kind regards,

Youssef El Khatib, Ph.D.

Academic Editor

PLOS ONE
---

## [Editor Report · Acceptance letter]

PONE-D-24-43200R1

PLOS ONE

Dear Dr. Gutierrez,

I'm pleased to inform you that your manuscript has been deemed suitable for publication in PLOS ONE. Congratulations! Your manuscript is now being handed over to our production team.

Kind regards,

on behalf of

Prof. Youssef El Khatib

Academic Editor

PLOS ONE